# Clinical and Research Solutions to Manage Obstructive Sleep Apnea: A Review

**DOI:** 10.3390/s21051784

**Published:** 2021-03-04

**Authors:** Fen Xia, Mohamad Sawan

**Affiliations:** 1Zhejiang University, Hangzhou 310024, China; xiafen@westlake.edu.cn; 2CenBRAIN Laboratory, School of Engineering, Westlake University, Hangzhou 310024, China

**Keywords:** obstructive sleep apnea, continuous positive airway pressure, oral appliance, weight loss, OSA detection and treatment, hypoglossal nerve stimulation (HGNS), electrical stimulation

## Abstract

Obstructive sleep apnea (OSA), a common sleep disorder disease, affects millions of people. Without appropriate treatment, this disease can provoke several health-related risks including stroke and sudden death. A variety of treatments have been introduced to relieve OSA. The main present clinical treatments and undertaken research activities to improve the success rate of OSA were covered in this paper. Additionally, guidelines on choosing a suitable treatment based on scientific evidence and objective comparison were provided. This review paper specifically elaborated the clinically offered managements as well as the research activities to better treat OSA. We analyzed the methodology of each diagnostic and treatment method, the success rate, and the economic burden on the world. This review paper provided an evidence-based comparison of each treatment to guide patients and physicians, but there are some limitations that would affect the comparison result. Future research should consider the consistent follow-up period and a sufficient number of samples. With the development of implantable medical devices, hypoglossal nerve stimulation systems will be designed to be smart and miniature and one of the potential upcoming research topics. The transcutaneous electrical stimulation as a non-invasive potential treatment would be further investigated in a clinical setting. Meanwhile, no treatment can cure OSA due to the complicated etiology. To maximize the treatment success of OSA, a multidisciplinary and integrated management would be considered in the future.

## 1. Introduction

Sleep apnea is a highly prevalent sleep disorder that is characterized as repeated reduction or cessation of airflow because of upper airway resistance and pharyngeal collapsibility during sleep [1,2]. Patients who suffer from apnea show various symptoms that include excessive daytime sleepiness, loud snoring, sleep complaints (e.g., hypersomnia and insomnia), nocturnal cerebral hypoxia, morning headache, and nocturia [3,4].

The earliest report of periodic breathing in sleep can be traced back to the mid-1850s and the obstructed sleep apnea (OSA) was first reported in the 1870s [5]. However, we began to have a deep understanding of the etiology and pathogenesis of apnea only from the mid-1950s [6]. According to pathogenesis, sleep apnea can be classified as OSA, central sleep apnea, and mixed apnea. OSA, the most common type, is an otorhinolaryngological problem characterized by the recurring occlusion of the upper airway during sleep. Central sleep apnea is a neurological problem caused by the ability of the neural function to maintain the upper airway patency needed to relax during sleep. Mixed apnea is an integration of central and obstructive apneas [7]. Neural apnea occurs less compared with OSA. In this review paper, we focus on the treatments of obstructive sleep apnea.

OSA has a significant adverse influence on patients’ health-related quality of life that increases morbidity and mortality and burdens social and psychological implications [8,9]. There is a close association between untreated OSA and high risk for the incidence of a series of hypertension, cardiovascular disease, metabolic disorders, and cognitive impairment [10]. The increase of morning blood pressure is linear to the severity of OSA regardless of age and Body mass index (BMI) [11]. Individuals with moderate-to-severe OSA have a 2-fold increased risk of stroke [12]. OSA even has a potential association with emerging pandemic as it may increase the death risk from COVID-19 [13].

The gold standard treatment of moderate-to-severe OSA patients is CPAP. However, almost 50% of patients cannot tolerate this treatment in the long term. Other non-invasive treatments such as using an Oral appliance (OA) and losing weight have been recommended, but these alternatives still have a low success rate of apnea-hypopnea index (AHI) less than 5. Surgical modification is suitable to select patients especially for those who are unable to tolerate CPAP therapy [8]. Surgical approaches include MMA, uvulopalatopharyngoplasty (UPPP), tonsillectomy (TE), and adenotonsillectomy (AT). MMA is a complex surgical advancement by advancing the facial skeletal framework surgically to enlarge the pharyngeal and hypopharyngeal airway dimensions [14]. UPPP procedure is a palatopharyngeal surgery that consists of bilateral tonsillectomy and partial amputation of the uvula [15]. Though each treatment can alleviate the symptoms to some extent, it has its limitations and side effects. 

Neural electrical stimulation has recently provided an alternative option, as it can keep the upper airway open when stimulating the hypoglossal roots during the inspiration phase [16]. Emerging neural electrical stimulation-based solutions aim to propose both the portability of a high-performance device and viable long-term therapy [17,18,19].

The aim of this review paper is to identify practicable medical solutions to treat OSA, review the common approaches and summarize their methodologies, illustrate the latest research on treatments, and evaluate the corresponding success rates and limitations. The remainder of this paper includes the analysis of the diagnoses and burden in Section 2. In Section 3, we describe the available treatments and compare their advantages and drawbacks. The final section concludes future research directions. 

### Methods

Google Scholar, PubMed, Web of Science, and IEEE explore databases were used to search and find the prior-art publications focusing within the 2000–2021 window. The materials for this review paper were found by keywords provided on the front page of this paper. We mainly focus on the selection of randomized controlled trials, review, systematic reviews, meta-analysis, and clinical trial journals written in English. Due to the large quantity of papers, we mainly sorted out the main contributions published during the last five years. Finally, 128 published articles were included in this review.

## 2. Diagnoses and Burden

OSA is characterized by recurrent cessations or reductions of airflow in the upper airway during sleep. The oral pharayngeal region is the normal site of airway collapse. The narrow upper airway is a common phenomenon which is caused by abnormal fat deposition or abnormal cranial bony structures in most OSA subjects. These abnormalities include shorter length of mandibular body, inferior positioning of hyoid bone, retrognathia, longer soft palates, narrower hard palates, and wider uvulas [20,21]. Obesity is regarded as the major risk factor for OSA. Besides obesity, the prevalence of OSA increases in the elderly population [22]. 

OSA presents a tremendous threat to the global healthcare system affecting approximately one billion people worldwide (one-seventh of the world population) [23]. In 2015, the cost of OSA diagnosis and treatment in the United States was more than 12.4 billion dollars for 2.5 million patients, while the estimated cost for the 23.5 million undiagnosed patients with OSA is 150 billion dollars [24]. In 2019, the number of OSA patients aged 30–69 years worldwide with AHI ≥ 5 was nearly 1 billion [24,25]. This huge number of OSA patients will significantly increase the economic burden globally. Meanwhile, the large number of OSA patients could not be treated promptly because appropriate management services remain unavailable [26].

For OSA diagnosis, polysomnography (PSG) is a standard laboratory-based test which monitors the sleep and respiratory parameters overnight. Classic obstructive apnea is accompanied by the absence of airflow, the presentence of arousal from polysomnogram, and the decrease of oxygen saturation [27]. AHI is a main parameter for assessing the severity of OSA that is derived from PSG results. AHI defines the number of apneas and hypopneas per hour while asleep [28]. 

However, the AHI-based standard diagnostic is different in adults and children. For adults, AHI ≥ 5 and <15 indicate mild OSA, AHI ≥ 15 and <30 moderate, and AHI ≥ 30 severe [29]. However, an AHI above 1 means mild OSA among children [21]. In this review, the treatment success is defined as the reduction of the AHI index to less than 5 or more than 50% reduction of AHI according to the American Academy of Sleep Medicine [30].

Questionnaires are available to screen and assess OSA risk in the primary care setting for the advantages of simple operation and low labor cost. Commonly used assessment questionnaires for sleep disorders include: (1) Berlin questionnaire, which is an assessment tool based on 10 questions about snoring behavior, daytime sleepiness or fatigue, and presence of obesity or hypertension to identify the sleep apnea risk in the primary healthcare settings [31]; (2) STOP-Bang questionnaire (SBQ) a screening for preoperative assessment, eight dichotomous items (yes/no): snoring, tiredness, observed apnea, high blood pressure, body mass index, age, neck circumference, and male gender) [32]; and (3) NoSAS is used to assess by medical records reviews which include neck circumference, body mass index, snoring, age and gender) [33], and Epworth sleepiness scale (ESS, the most widely used questionnaire to assess daytime sleepiness, and low sensitivity for OSA [34]. To discriminate the severity of OSA, NoSAS showed better performance than SBQ [35]. 

Home sleep apnea testing is increasingly widespread to be used for screening OSA. It is an unattended portable monitor that needs no in-laboratory attendant. Individuals can apply the monitor at home and follow the instructions of a technician via video. The sensitivity of this test is slightly lower than PSG, but it saves much labor cost and brings much convenience and comfort to the patients [36,37].

## 3. Treatment of OSA

As the number of patients with OSA continues to increase, a series of treatments has been put forward. In this section, we review the most commonly used treatments and elaborate on contemporary studies of invasive and non-invasive methods. Management of OSA includes CPAP, oral appliance therapy, behavioral modification, surgical procedures, as well as electrical stimulation.

### 3.1. CPAP

Multiple treatments are available depending on the severity and pathogenesis of OSA, physicians’ suggestions, and patients’ preferences. The most widely used method for treating OSA is CPAP. A CPAP system provides constant pressure to the airway through a tube connected to a face mask or nasal mask. It provides a continuous flow of inspired gases to keep airways firm and open and prevent collapse of airway during inspiration [38]. The therapeutic effect is related to the duration of CPAP treatment as well as the adherence of the OSA patients. The minimal use of CPAP is recommended as 4 hours per day and 5 days per week according to acceptable adherence of clinical definition [39].

The use of CPAP could reduce the comorbidities of cardiovascular disease, hypertension, and diabetes while improving the quality of life [40]. It also reduces blood pressure for patients with cardiovascular diseases and multiple cardiovascular risks in the middle-aged population [41]. Quality of life encompasses health-related complaints, physical fitness, psychological-social functioning, and emotional status [42]. Through CPAP treatment with a duration of at least two weeks, which cannot alleviate the impaired psychological status, it helps to improve the physical symptoms [43]. The therapeutic effects of this treatment on improving impaired cognitive performance of OSA’s patients is related to the duration of CPAP [44].

The greater adherence to the CPAP treatment is reflected when participants show higher improvement on sleepiness and quality of life [9]. Three-month treatment with CPAP benefits blood pressure and metabolic syndrome of severe OSA in elderly patients [45]. Recent research outcomes have revealed that CPAP treatment is beneficial for visual sensitivity and retinal thickness in OSA patients [46]. Except for those who need supplemental oxygen, there is an improvement in the outcome of interstitial lung disease with the help of CPAP regardless of OSA severity and adherence [47]. A 2020 data analysis of 55 moderate to severe OSA patients who underwent nocturnal PSG indicated that there were improvements in heart rate variability index after one night of CPAP therapy. It was suggested that these improvements would reduce the incurrence risk of cardiovascular disease [48].

However, in the elderly population (higher than 65 years old), the benefit of CPAP on sleepiness and health-related quality of life is obscure [49]. Although CPAP is expected to be helpful for the prevention of cardiovascular events in OSA patients, current data reveals that whether it improves cardiovascular events such as stroke and unstable angina is unclear [50]. Even patients with OSA had been treated by CPAP for 12 or 24 weeks, the glycated hemoglobin showed no improvement [51]. For stroke-affected OSA patients, there is a significant improvement in the attention and executive function after CPAP improvement, but no clear improvement in the neurocognitive function or quality of life [52]. 

The classical side effects of CPAP include nasal dripping, rhinitis, nasal discomfort, and mucosal drying which may contribute to poor adherence. A retrospective study of OSA patients who underwent CPAP therapy showed that CPAP compliers (>4 h/night) accounted for 59.3% during 7 years follow-up [53]. The termination of CPAP is more likely in females versus males, in patients who receive treatment in public versus personalized management [54]. Improving CPAP adherence is necessary to maximize the efficiency of CPAP management. Promising factors that are useful to improve patient’s self-efficacy of CPAP adherence include cognitive behavioral therapy, motivational enhancement therapy as well as education about the risk of OSA [39,55]. 

### 3.2. Oral Appliances 

An OA is a mouth guard or retainer that can enlarge the upper airway and reduce pharyngeal collapse by changing the position of the lower jaw into an anterior area. This non-invasive and custom-fitted device lets the patient feel as comfortable as possible. It is increasingly becoming an alternative treatment for OSA as it is more accepted by patients compared with CPAP. Among various intraoral devices with different action sites, mandibular advancement devices are the most commonly used for treatment of OSA [56]. 92% of patients with OSA can benefit from OAs for improving the AHI index and symptoms of OSA [57]. 

The treatment success of OAs in 425 OSA patients was 68% related to AHI scores regardless of demographic and anthropometric factors [58]. The treatment success rate of OAs has a strong relationship with the reposition distance of the lower jaw. The larger distance of the lower jaw put forwards, the bigger reduction of AHI. But the advancement higher than 50%, success rate did not obviously increase [59]. Among two mandibular advancement device configurations: a one-piece (monobloc) or two-pieces (duobloc), the monobloc showed a higher success rating of 82.1% than the duobloc (52.1%) from metanalysis [60]. An adjustable OA has a higher success rate than a fixed one. People that are younger, have a lower BMI, and suffer less severe diseases are more successfully respond to this treatment [61]. Generally, custom-made devices provide better comfort and efficacy than non-individualized OA devices in patients with OSA [62]. The mild-to-moderate OSA patients have a greater chance of success compared with severe OSA [63]. Severe OSA patients are more likely to be recommended CPAP therapy in clinical. For CPAP-failed patients with severe OSA, OAs are also successful for AHI reduction [64].

Among patients with OSA, both CPAP and OA therapies were related to blood pressure reduction with 4-week follow-up, but there is a lack of metanalysis to identify the difference of blood pressure outcomes between these therapies [65]. To further evaluate the outcome of blood pressure and endothelial function, research on individuals with mild OSA has been done. After 1-year treatment of CPAP or OAs, even patients have successful treatment, the improvement to blood pressure or arterial function did not explicit from the newest paper [66]. 

A 2015 randomized-control-trial of 40 OSA patients who underwent mandibular advancement devices and CPAP therapy revealed that both interventions had beneficial changes in cardiac function including heart rate variability and blood pressure after 12-week treatment [67]. Further meta-analysis for the OA therapy in cardiovascular effects showed that therapy may be associated with long-term reduction in cardiovascular morbidity and mortality in OSA patients, but this speculation needs large databases and high-quality methodology to verify [68]. 

The most side-effects of long-term OA therapy are dental movement and skeletal changes which will result in a decrease in overjet and overbite. The reciprocal forces on teeth and jaw exerted by the OA device bring these mechanical side effects. At some point in the future, patients may be disturbed by esthetics or chewing or biting problems [69]. Dental changes are severer than skeletal changes with long-term OA therapy in OSA patients [70]. Minor temporary adverse effects include pain, excessive salivation problems, mouth dryness, and discomfort. These temporary side effects will disappear after few months generally. The risk of developing pain and structural impairments hampered the long-term use of the OA device [71]. In recent years, a salivary biosensor was developed to be potentially useful for monitorization of patients who wear an OA [72].

Even CPAP is more efficient to reduce AHI, OA therapy is more favorable in OSA patients. Through 12 months monitoring 59 moderate OSA patients, the adherence of OA in self-reported with questionnaire was higher than CPAP, but objective data revealed that both therapies have comparable adherence [73]. Neither method showed significant improvement in quality of life, cognitive, or functional outcomes in OSA patients through metanalysis [74]. A positive airway pressure device combined with OA provided higher treatment efficiency in reducing the severity of OSA compared with OA alone. Among 22 individuals with OSA, nine patients resolved the symptom in the AHI index (AHI <5) with a positive airway pressure device combined with OA management [75]. A one-year follow-up showed that mandibular advancement therapy was less cost-effective and clinically effective in terms of AHI than CPAP in moderate patients [76].

### 3.3. Weight Loss

As noted above, obesity is a high risk for OSA and at least 70% individuals with morbid obesity (BMI > 40 kg/m^2^) have OSA [77]. Increased fat deposits around the pharynx are regarded as a key mechanism through which obesity results in OSA. Peripharyngeal fat deposition can produce an extra-mechanical load that can prevent dilator muscles from keeping the airway patency. Through weight loss, nasopharyngeal collapsibility can be reduced, thereby increasing the capability of the upper airway. The pharyngeal fat pad area may play a crucial role in the early stage in overweight patients with OSA. Weight reduction through behavior intervention brings an improvement in obese OSA patients [78].

Patients who lose more weight and those with mild OSA initially are more likely to be cured. Five to 10% weight loss shows an improvement of OSA severity. To achieve significant improvement in OSA, minimal 7% to 10% weight loss should be recommended clinically. In fact, weight loss of 25 to 30% makes the best performance for AHI reduction [79]. The methods to reduce weight include diet intervention, moderate physical exercise, as well as bariatric surgery. 

A random-controlled test among 702 participants with OSA investigated that both diet intervention and exercise were associated with the reduction of AHI, only diet intervention was significantly efficient in BMI reduction [80]. Adopting a strict low-calorie diet can reduce the mean BMI by 4.8 kg/m2 and the mean AHI by 14.3 [81]. Twelve-week high-intensity interval training showed a positive effect on AHI and daytime sleepiness even if there was no improvement in BMI. Through exercise intervention, AHI reduced by 7.5 ± 11.6, and sleepiness that was assessed by the Epworth scale questionnaire improved from 10.0 ± 3.6 to 7.3 ± 3.7 [82]. A prospective multicenter trial among 132 OSA patients showed that the prevalence of OSA decreased by 27% after bariatric surgery [83].

There are multiple connections among OSA, obesity, and disturbed glucose homeostasis. Treating each one is beneficial for the improvement of others [84]. Weight loss in patients with type 2 diabetes has a positive effect in AHI reduction. Through an intensive life intervention, the reduction of AHI was bigger by intensive life intervention than only diabetes supports and education [85]. Weight loss by bariatric surgery in morbid obese OSA patients had a beneficial effect on the reduction of adverse cardiovascular events [86].

Weight loss is efficient for alleviating the severity of OSA but is scant to cure. Through bariatric surgery on patients with OSA, the mean BMI reduced from 51.0 kg/m^2^ to 32.1 kg/m^2^ and the mean AHI score reduced from 47.9 to 24.5. At 1 year follow-up, 71% of patients still had moderate or severe disease after weight loss treatment that required CPAP therapy and only 4% of patients had normalized OSA [87]. A promising proportion (22%) of obese patients with nonsupine OSA was cure with AHI less than 5 via weight loss [88]. Nevertheless, it is difficult to achieve and maintain weight loss by lifestyle intervention as the adaptive physiologic neurohormonal changes in response to weight reduction. A two-year program showed that there was around 30% drop-out [89]. The cost of a weight loss program is complex, as it involves the employment of a dietician, personal trainer, and physiotherapist as well as nurses. Developing a customized and multidisciplinary weight management should be targeted in OSA patients with morbid obese [90]. 

In addition to weight loss, regular breathing retraining activities such as singing and playing wind instruments play a potential role in OSA treatment especially for mild OSA patients [91].

### 3.4. Surgical Procedure

Although CPAP is regarded as a gold standard treatment for OSA patients, there is low compliance of sticking to this device because of pathophysiological factors, especially airway obstruction in the nose, palate, tonsil, uvula, tongue, and pharynx. Because of this, upper airway surgery is an alternative for those with severe OSA. Figure 1 summarizes the procedure for choosing a suitable surgery treatment. Before airway surgery, a complete sleep history is recorded and a physical exam that includes neck circumstance, BMI, and facial skeletal character is carried out. Then, a sleep specialist will assess the severity of OSA using nocturnal PSG data. Other treatments in Figure 1 involve CPAP, OA, behavior intervention, and pharmaceutical.

For mild OSA patients, non-invasive treatment is suggested as the first-line management. Anti-inflammatory therapy showed good performance in normalization of pediatric mild OSA with 62% treatment success after 12-week treatment [92]. CPAP or OA or behavior intervention is useful for adult OSA [91,93].

For severe OSA patients, a lateral cephalometric radiograph combined with fiberoptic nasopharyngoscopy is conducted to check whether the palate is redundant, oropharyngeal airway is crowded, or skeletal structure is abnormal. Teeth marks on each side of the tongue indicate macroglossia. From the facial skeletal character and lateral cephalometric radiographs, the maxillomandibular deficiency can be identified. Adult OSA patients with a nasal obstruction who have a nasal structure abnormality would have a septoplasty or turbinectomy operation to enlarge the nasal airway. For normal nasal structure, CPAP or OA can be used for the primary treatment.

For severe OSA patients diagnosed with a long uvula and redundant soft palate, UPPP is a common method to enlarge the retropalatal airway based on the advice of a specialist. The UPPP post-operational results show that snoring, daytime sleepiness, and the oxygen desaturation index improve to different degrees. As invasive management, the UPPP procedure involves complications including velopharyngeal insufficiency, dysphagia, post-operative bleeding, swallowing difficulty, and nasopharyngeal reflux fluid after the operation. The success and response rate of UPPP with long-term longer than 34 months was less effective than short-term between 3 months and 12 months (44.35% vs. 67.3%) [94]. A 6-month follow-up after UPPP surgery showed that there was a significant positive effect in cardiac parameters [95]. To predict the treatment success after UPPP, Friedman stage I based on tonsil size and palate position shows better performance over age, preoperative AHI, and other physical parameters [96]. For OSA patients with unsuccessful UPPP surgery, OA therapy is an effective alternative. Almost 50% of patients achieved a normal AHI index that less than 5 and 73% of patients had a 50% reduction of AHI via OA management [97]. Compared with UPPP surgery alone, the combination of UPPP and tonsillectomy (TE) showed a higher treatment success. In a controlled trial, almost all patients were satisfied with the outcome of UPPP and TE intervention and 65% of patients had been cured from OSA after that needed no further treatment [98]. CPAP is not an efficient therapy for OSA patients who had failed to respond to UPPP and TE management. Pointing to these patients, upper airway stimulation would be an adjunct solution [99].

Maxillomandibular advancement (MMA), a classical efficient treatment for patients with a maxillomandibular deficiency, has a 100% surgical success according to the scores of AHI and respiratory disturbance index [100]. There are two methods of maxillomandibular advancement: surgery and orthodontic device use. In this section, we categorize MMA as a surgical treatment consisting of the surgical advancement of the distal segment, sagittal ramous split osteotomy, and fixation appliance. Younger and healthier OSA patients are more likely to be cured by MMA [101]. Lateral pharyngeal wall collapsibility was improved after MMA surgery that was associated with treatment success. The mean pharyngeal airway volume and pharyngeal airway space have been increased in OSA patients after MMA intervention [102]. MMA is a highly invasive surgical procedure with many complications including malocclusion, hemorrhage, local infection, facial numbness, pain, swelling, tingling, chin stiffness, poor cosmetic result, and even postsurgical relapse of advancement. Most patients can recover regular function within 2 to 10 weeks after MMA surgery [103]. 

For OSA patients who have cardiac or pulmonary disease, the treatment should be considered seriously. CPAP can be recommended as the primary treatment. For patients who failed with CPAP, the conventional surgery including UPPP and MMA may be a good option. But there are some criteria for conventional surgery, if the patients cannot meet the criteria or refused to accept the complex surgery procedure, upper airway stimulation may be an alternative [104].

For children with OSA, the primary treatment is adenotonsillectomy (AT) to address adenotonsillar hypertrophy which is the most etiology of pediatric OSA [105]. However, over half of children had residual OSA after adenotonsillectomy surgery. AT performed a significant AHI reduction in pediatric OSA from 18.2 ± 21.4 to 4.1 ± 6.4 and 27.2% cured after AT surgery with AHI less than 1 [106]. Children with higher BMI, severer OSA, and craniofacial abnormality are more likely to persist OSA symptoms [107]. Improvement in cognitive function has not been found with OSA children in a year after adenotonsillectomy surgery [108]. CPAP, weight reduction, and OA are potentially useful for poor outcome AT candidates. Moreover, new surgical procedures and techniques including mini-invasive septoturbinoplasty and palatal suspension instead of excision are desirable [109]. To improve long-term treatment success of pediatric OSA, a multi-discipline approach will be preferred in future research [110]. 

### 3.5. Electrical Stimulation

In 1996, scientists found that pharyngeal collapse at the beginning of sleep is strongly correlated with the loss of genioglossal muscle tone [111]. The genioglossal muscle, the largest dilator muscle that maintains upper airway patency, can be contracted by the electrical stimulation of the hypoglossal nerve to enlarge the upper airway and reduce the AHI. The HGNS scheme used for OSA treatment dates back to 1997 [112]. Later, Schwartz put forward that HGNS could reduce the AHI and frequency of respiratory interruption as well as improve oxyhemoglobin saturation during sleep [113]. The system mainly consists of three implantable parts: a respiratory sensing lead, stimuli generator, and stimulating electrode. The respiratory sensing lead detects the pattern of breathing by sensing the bio-impedance with the chest wall motion and transmits the information to the stimuli generator. Then, the latter provides the electrical stimulation pulses to the stimulating electrode that delivers this stimulation signal to the hypoglossal nerve. The flowchart is summarized in Figure 2, where *t* is the average duration time of the practical stimulation and *n* is the preset stimulation duration. The first step of the simulation process is to predefine the initial state of the stimulation duration; here, we suppose the preset stimulation duration is *n*0 (hours or minutes) and the practical stimulation starting time is 0. Then, the sensing task consists of monitoring the respiratory rhythm to assess whether the patient is in the inspiration phase. If the patient is in the inspiration phase, the sensor triggers the signal generator that delivers the electrical stimuli. Using the stimulating electrode, the electrical stimuli are sent to the hypoglossal nerve. However, if the patient is not in the inspiration phase, the sensor does not activate the signal generator and must wait for the patient’s forthcoming inspiration phase. 

The cycle of one stimulating process takes T0 minutes, and n cycles make the practical time t equal to *n* × T0. The stimulation process ends when the value of practical time t is no less than the preset stimulation duration *n*0. However, there have been a series of technical problems including sensing sensitivity, the failure of the stimulating electrode, and the stimulating electrode lead being in contact with tissues, which all affect the development of HGNS. 

There are strict exclusion criteria before HGNS surgery including BMI > 32 kg/m^2^, neuromuscular disease, obstructive pulmonary disease, severe hypertension, and cardiovascular disease. Participants also need to undergo in-laboratory screening tests including PSG, drug-induced sedated endoscopy, and surgical consultant assessment [114]. In a retrospective analysis, there was no device-device interactions between transvenous cardiac implantable electronic device and implantable hypoglossal nerve upper airway stimulation in the early experience [115]. HGNS showed a high surgical success rate of 76.9% with ImThera device. After 12 months, there was still a high mean AHI reduction of 24.2. HGNS showed high efficacy and safe performance in OSA patients not tolerating or showing no response to CPAP therapy [116]. Statistically, the factors most affecting the improvement of post-operational outcomes include the original AHI, patient age, and BMI. Higher AHI, older, and lower BMI patients show a greater reduction in the AHI after HGNS surgery [117].

A meta-analysis among 837 OSA patients (517 patients were treated by positive airway pressure, 320 patients with upper airway stimulation implantation) showed that positive airway pressure had more improvement in diastolic blood pressure compared with upper airway stimulation (3.7 mm Hg vs. 2.8 mm Hg). The adherence of upper airway stimulation was better than positive airway pressure [118]. Outcome comparison of HGNS and conventional surgery UPPP in OSA patients showed that mean AHI reduction from 38.9 ± 12.5 to 4.5 ± 4.8 with HGNS and 40.3 ± 12.4 to 28.8 ± 25.4 with UPPP. HGNS performed ability of normalizing AHI index (AHI < 5) in the majority of patients [119]. HGNS provides a potent option for severe OSA patients with dentofacial deformity who refuse facial skeletal surgery [104].

Although non-serious adverse effects occur during the HGNS process, there exist common adverse events included pain, tongue abrasion, device malfunction, abnormal sensations, paresthesia, change in salivary flow, and lip weakness [120].

More recently, various electrical stimulation devices have been introduced for research purposes. Common sponsors include Inspire Medical Systems Inc., Apnex Medical Inc., and ImThera Medical Inc. The overall constitution of these stimulators is the same. The subtle difference is in the number of sensing electrode leads and their shapes [121]. The newest ImThera Aura6000 system includes a rechargeable battery and a lower volume of the stimulator. In addition, a tiny device, the Nyxoah system, is placed on the top of the genioglossus muscle to stimulate the hypoglossal nerve to help cure OSA symptoms. The implantable stimulator is a simple circuit powered by a coil that converts electromagnetic energy from the external device into electrical energy; this reduces the overall volume of the stimulator [122].

The transcutaneous electrical stimulation (TES) is another potential effect and safe stimulation to treat patients with OSA. Two adhesive electrode patches were placed onto the skin of the submental area which is halfway between the chin and the angle of the mental region. With a little battery power, transcutaneous electrical stimulation was delivered to the submental skin to activate the motor unit of the dilators of the upper airway by stimulating the underlying nerve [123]. In a randomized, sham-controlled trial, the mean AHI after one-night transcutaneous electrical stimulation was reduced by 9.1 from 28.1 while the sham stimulation had a modest reduction of AHI by 4.1 from 19.5 [124]. In a paper [125] that compared the treatment effect of HGNS and TES on patients with OSA, AHI reduced by 24.9 in HGNS intervention and TES reduced by 16.5 in TES treatment. TES showed less treatment efficiency than HGNS, but it was much more cost-effective and had almost no adverse events besides minor local skin irritation. TES had a similar stimulation methodology with HGNS that it would be used for predicting the treatment success of HGNS in patients with OSA before undergoing an invasive surgery [123]. Moreover, the TES is a non-invasive therapy that avoids complex surgical procedures, and the proof-of-concept outcome for patients with OSA is promising, even it has not been implemented into clinical management yet. To apply TES in a clinical setting, patient selection is very important that is highly related with the treatment failure. From [121], responders to TES prior have the AHI less than 20 before this treatment.

It is worth noting that the vagus nerve stimulator as a treatment for drug-resistant epileptic seizures associated with severe OSA. However, the vagus nerve stimulator is not for treating OSA but inducing severe OSA [126].

## 4. Comparison of Each Treatment

Despite the scarcity of studies comparing OSA treatments, we extracted the results from the main published contributions and compared cure rates. Table 1 shows the comparison of success rates and limitations of different treatments. The treatment success criteria are defined as AHI index to <5 or more than 50% reduction of AHI in adults with OSA. For children with OSA, AHI less than 1 or reduction of AHI higher than 50% can be regarded as treatment success.

Treatments can be classified as non-invasive and invasive. CPAP, OAs, and weight loss are non-invasive methods. CPAP as the first-line OSA treatment showed a high effective outcome with a 59.3% treatment success rate. The mean AHI had a significant reduction with CPAP compliers by 32.9 from 48.6 in 7 years follow-up [53]. OA is always recommended as a secondary treatment in clinical and plays a more efficient role in less-severe OSA patients. With a customized OSA device, patients showed high treatment success. At some extent, the effect of OA can be comparable with CPAP. In a study of oral appliances, 68% of patients with OSA experienced treatment success in 4 years follow-up. Mean AHI reduced from 27.5 (before treatment) to 12 (after oral advancement device intervention) [58]. 

Comparing the effect and cost of CPAP and OA, CPAP was more cost-effective and clinically effective in patients with moderate OSA [76]. The main side-effect of OA is overjet and overbite after long-term management. Even weight loss management has a positive effect on OSA patients, but the treatment success rate is only 27% after 1 year of weight loss treatment [82]. Meanwhile, the maintenance of weight reduction is difficult, and the cost of weight management is complex. The combination of CPAP and weight loss is more useful for lowering blood pressure, insulin resistance, and lipid levels than any single method [127]. 

For OSA patients with no response to CPAP or oral appliance, invasive surgery would be a suitable option. Conventional UPPP and MMA showed great effective performance. The success and response rate of UPPP in long term was less effective than short term. MMA is a highly invasive procedure commonly used for treatment of maxillomandibular deficiency. It has a series of side effects including malocclusion, hemorrhage, local infection, etc. The treatment outcome of MMA also decreases with time. The study of 12.5 ± 3.5 years follow-up after MMA intervention showed the mean AHI reduced from 36.7 to 4.7 in success group and the treatment success rate was considerable with 27.6%. In this trial, there was a failure group in which patients had not been treated successfully via MMA and a success group in which patients had a good response to MMA surgery. Here we reviewed the success group data and marked in (s) [101]. The long-term outcome of UPPP was 44.35% after longer 34 months follow-up analysis. The mean AHI reduced from 39.9 to 21.5 among 215 severe OSA patients [94]. UPPP combined with TE showed higher treatment outcome than UPPP alone. The treatment success rate of 3 months after UPPP and TE was 64.5% [98]. From Table 1, we can easily observe that the AHI reduction after MMA was more pronounced than UPPP and UPPP and TE even the observation time of post-MMA was extremely long with 12.5 ± 3.5 years. 

For OSA patients with no response to UPPP surgery, OA therapy is an effective option. After failure of UPPP management, almost 50% of patients achieved a normal AHI index with OA [97]. For OSA patients who had failed to respond to UPPP and TE intervention, upper airway stimulation would be a great solution as CPAP has no efficient outcome [99]. HGNS, a relatively innovative method, displays the optimal therapeutic effect with a treatment success rate of 76% [117]. Generally, those who fail to benefit from UPPP surgery have a greater possibility of being unable to tolerate CPAP treatment [117]. The study showed HGNS had a better outcome in terms of more improvement after CPAP. The treatment outcome of HGNS was better than UPPP related to AHI scores [119]. There was a successful HGNS case in a severe OSA patient who had dentofacial deformity and HGNS provides a potent option for patients who have no response to MMA [104]. Here, we should explain that we did not show TES outcome here as the sample in clinical was sparse, but it does not mean the TES would not be a good alternative treatment for patients with OSA in the future.

For pediatric OSA, the primary treatment is AT even the treatment success is not high. After AT, the mean AHI reduced from 18.2 to 4.1 among 578 OSA children, and 27.2% of patients did not need to receive other treatment [106].

In terms of limitations, although CPAP is a mature treatment, around 50% of OSA patients fail to adhere to its standard usage. The limitation of OAs is the requirement of a good structure and need to be custom-fitted. From this perspective, weight management seems to have slight limitations except maintenance difficulty. The common complication of invasive treatment is pain, bleeding, and high infection susceptibility. Some patients have other post-operative complications including a decline in neurosensory and muscle soreness. Highly invasive MMA surgery has a higher risk of side effects than other OSA surgery treatments.

This treatment comparison has some limitations because of the lack of uniformity of different studies. The first is the different physical conditions of the OSA patients in each study because the screening criteria differ. Differences in demographic structure (gender, age, ethnic group, and region) are another important factor that influences the accuracy of the results significantly. The sample size, which affects the cure rate, in each study is not uniform. The cure rate of each treatment is also strongly related to the follow-up time. It is difficult to ensure the consistency of the follow-up time in different studies. The outcome of each treatment should consider not only the cure rate, but also other factors such as the effect of blood pressure, quality of life, sleepiness, and cardiovascular event risk. Despite these limitations, this comparison provides a guide for doctors and patients when choosing an OSA treatment.

## 5. Conclusions

OSA is a serious, potentially health-threatening disease that troubles millions of people each year. Clear guidelines from different studies and reports are lacking on how to decide on a suitable treatment for patients with different OSA severity. The current gold standard of OSA treatment is CPAP, as it can feed a positive air–oxygen mixture into the patient’s airway through the nasal obstruction continuously. However, studies find that OA is comparable with the CPAP method for non-severe OSA patients. CPAP combined with OA performs better outcome than CPAP alone and OA alone. For OSA patients with a higher BMI, weight loss is a long-term suggestion. The treatment efficiency of CPAP combined with weight loss is better than CPAP alone. Among invasive treatments of OSA, UPPP combined with TE has a higher cure rate than UPPP and MMA is a highly efficient management. OA is an efficient treatment for patients who failed to respond to UPPP intervention. However, UPPP, UPPP + TE, and MMA are irreversible interventions. HGNS, an emerging technique, displays favorable outcomes with a 76% cure rate and provides a strong alternative for those who did not respond to UPPP or MMA surgery. 

## 6. Future Research Directions

OSA is a life-threatening disease common worldwide. The sharp increase in the number of OSA patients each year is placing enormous financial strain on governments. A variety of OSA treatments has been put forward for relieving or curing the disease. However, research comparing these treatments is scarce. Therefore, developing a comprehensive guideline for choosing a suitable treatment for individuals with different severities of OSA is needed. To provide a comprehensive evidence-based comparison to guide patients and doctors, future research should consider the following:

Before treatment, specific screening and cure rate criteria should be formulated.

The comparison of these treatments should extend to the reduction of the AHI, complexity of the operation, cost, treatment time, and effect on physical function including quality of life and neurocognition.

Samples should be sufficient to relieve or remove the factor of the demographic structures of each part.

The preliminary research in this review paper shows that HGNS is a considerable prospect because of its high cure rate; it also has lower operational complexity than other surgical methods and does not need to remove redundant tissues. With the development of implantable medical devices based on a wireless power source and data transmission embedded in a smart electronic system in the human body for monitoring and treatment purposes [128]. HGNS will become smarter and more efficient in the future.

Another electrical stimulation treatment TES should be further investigated in a clinical setting as it can bring much more tolerance compared with HGNS and would be a potential non-invasive alternative for patients with OSA in the future.

To achieve efficient OSA therapeutic success, a multidisciplinary and integrated approach would be considered in the future study.

## Figures and Tables

**Figure 1 sensors-21-01784-f001:**
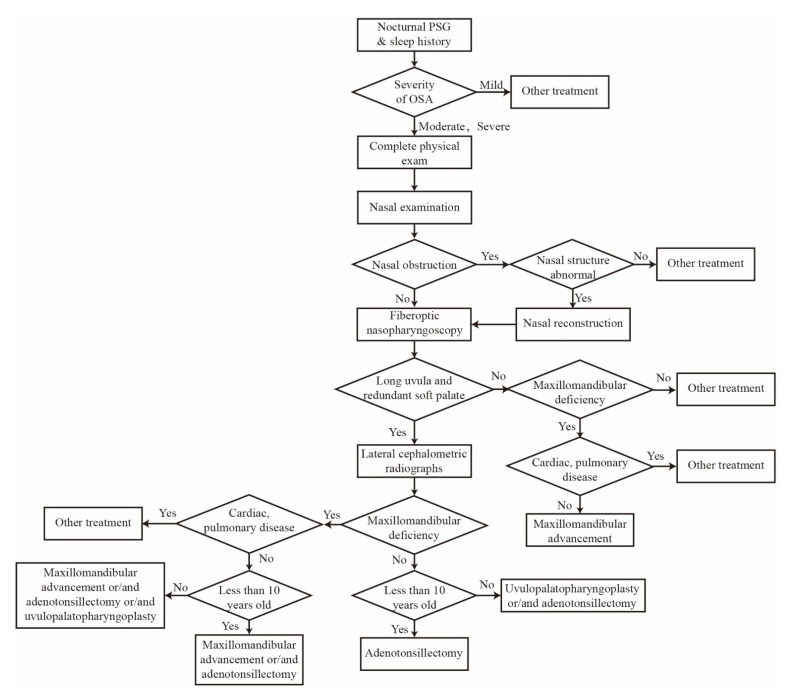
Potential procedure of surgery treatments option for obstructive sleep apnea patients.

**Figure 2 sensors-21-01784-f002:**
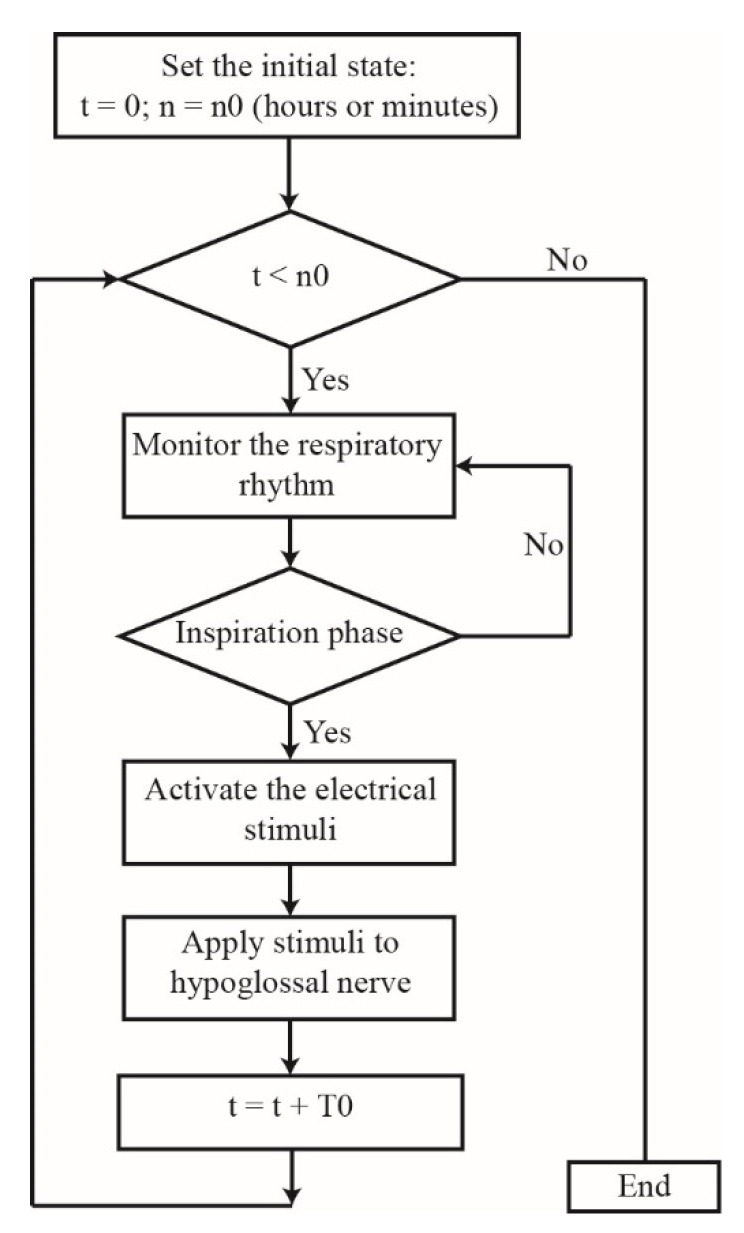
Hypoglossal nerve stimulation (HGNS) Procedure.

**Table 1 sensors-21-01784-t001:** Success rate and limitations of the OSA treatments.

	Methods	Non-Invasive Methods			Invasive Methods				
Limitation		CPAP [53]	OA [58]	Weight Loss [83]	MMA [101]	UPPP [94]	UPPP+TE [98]	AT [106]	HGNS [117]
**Severity of OSA**		Mild-severe	Mild-severe	Mild-severe	Severe	Moderate-severe	Moderate-severe	Moderate-severe	Severe
**Number of samples**		463	425	132	29	212	31	578	584
**Pre(mean)**		48.6 ± 31.8	27.5 ± 16.3	27.6 ± 24.6	36.7 ± 14 (S)	39.9 ± 18.3	33.7 ± 14.6	18.2 ± 21.4	33.8 ± 15.5
**Post(mean)**		5.7 ± 8.4	12 ± 12.5	9.9 ± 11.2	4.7 ± 3.2 (S)	21.5 ± 15.6	15.4 ± 14.1	4.1 ± 6.4	11 ± 13.6
**AHI < 5 or AHI reduction > 50%**		59.3%	68%	27%	27.6%	44.35%	64.5%		77.1%
**AHI < 1**								27.2%	
**Follow-up**		7 years	4 years	1 year	12.5 ± 3.5 years	≥34 months	3 months	Immediately	1 year
**Efficiency**		+++++	+++	+	+++++	+++	++++	++	+++++
**Limitations**		Poor adherence	Strict teeth structure, long-term overjet and overbite	Difficult to achieve weight loss and maintain	Highly invasive and complicated procedure, side effects include malocclusion, hemorrhage, facial numbness, etc.	Velopharyngeal insufficiency, dysphagia, swallow difficulty	Velopharyngeal insufficiency, dysphagia, swallow difficulty	Post-operative bleeding, infection of wound	High cost, tongue abrasion, device malfunction, abnormal sensations, etc.

CPAP, continuous positive airway pressure; OA, oral appliance; MMA, maxillomandibular advancement; UPPP, uvulopalatopharyngoplasty; AT, adenotonsillectomy; TE, tonsillectomy; OSA, obstructive sleep apnea; AHI, apnea-hypopnea index; HGNS, hypoglossal nerve stimulation; Pre(mean) is the mean AHI value before treatment; Post(mean) is the mean AHI value after treatment. More ‘+’ sign means the higher treatment outcome.

## Data Availability

Data sharing not applicable.

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
