# Peer review of "Clinical and Research Solutions to Manage Obstructive Sleep Apnea: A Review"

_sensors, 2021, doi:10.3390/s21051784_

Round 1
Reviewer 1 Report
I reviewed your manuscript entitled: " Clinical and Research Activities to Cure Obstructive Sleep Apnea: A Review” and found it relevant to the study of OSAS for the general practitioner, and as such of interest to the readers of the journal.
Some points of criticisms should be addressed before the manuscript can be considered for publication.
My comments are the following:
- The title has to be corrected from CURE OSA to MANAGE OSA since many of the modalities described did not really cure the syndrome.
- The authors are kindly requested to give more information regarding the search strategy including the MeSH keyword terms
- Section 4 (Studies of each treatment) has to be integrated in the text of section 3 according to the different treatments.
Author Response
We would like to thank you for your comments to improve our paper. We have fully revised our manuscript and will answer your comments one by one in the following.
Thank you once more in advance for your further consideration.

Reviewer 2 Report
Thank you for asking me to review this paper about treatments for sleep disordered breathing.
The topic is of interest for the readers of Sensors Journal however I don’t feel this work can be published due to the following reasons:
- Throughout the manuscript the authors mix the concept of treatment and cure of the disease. Cure usually refers to a complete restoration of health, while treatment refers to a process that leads to an improvement in health, but may not include the complete elimination of disease. OSA can be cured in few cases only, whilst in the vast majority it can be treated. In this paper most of the treatment are discussed but they don’t correspond to the cure of the disease
- The text is very difficult to read. English level is very low, I suggest a thorough review by a native English speaker
- The description of the disease is confusing: feature like somnambulism are described as frequent whilst other important symptoms like unrefreshing sleep, nicturia are not mentioned
- The authors state that they conducted a systematic search without stating the terms of the search (keywords, timeframe)
- “Questionnaire can also be used to diagnose OSA.” Questionnaires are not a diagnostic tool but rather a screening tool.
- “However, this does not mean that 128 using CPAP can stop the symptoms of OSA.” This sentence does not make sense
- The references added in the text do not seem up to date. Important publications of the last 5 year period are missing
- Figure 1 shows surgical treatment only. I this was done on purpose, caption should be changed
- The discussion about treatments for OSA seems unbalanced towards surgery and in particular Hypoglossal nerve stimulation.
- There is no mention about transcutaneous electrical stimulation which has proven effective in patients with OSA in a recent RCT (http://dx.doi.org/10.1136/thoraxjnl-2016-208691)
Author Response

(The authors gave the same response as above.)

Reviewer 3 Report
Manuscript entitled „ Clinical and Research Activities to Cure Obstructive Sleep Apnea: A Review” summarizes information on available treatment in OSA.
The topic of the review should be a meta-analysis considered available data, with additional comments on the articles that would not fulfill the criteria for inclusion in meta-analysis.
Abstract: HGNS abbreviation is not expanded
Methodology. If the review was systemic, authors should provide number of they found, on what bases were there searched for and further included, and if the studies were first screen on title bases or abstract. A figure summarizing the process would be helpful for the reader.
If authors decide to list all apneas, and further define only OSA and central apnea, there is a need to also define mix apnea.
Page 3, line 98/99: AHI includes not only apneas but also hypopneas, so it includes not only pauses in breathing but also periods of shallow breathingwith accompanied by decrease in oxygen blood saturation.
Page 3, line 103/105: While disusing OSA, it is not on topic to include most of sleep questionaries, as there as not designed for the diagnosis of sleep disordered breathing. What is more, short description on applicable questioners should be provided on which bases they can help with OSA diagnosis. Further, scales such as STOPbang or noSAS should also here described. It should be noted that they questionnaires and scales help in the diagnosis; they do not do it on their own.
CPAP abbreviation is not expanded the first time it appears in the manuscript.
Figure 1. Where the other treatment is put authors should point, which forms of treatment should be considered there.
The is a need for a language correction, grammatical improvements, some sentences need to be rephrased. Fe. “Sleep apnea is usually associated with a sleep condition that is characterized as pausing breathing during sleep„ (line 34/35, page 1). Sleep apnea is the condition that is characterized by pauses in breathing during sleep, it is not associated with sleep condition.
Author Response

(The authors gave the same response as above.)

Round 2
Reviewer 2 Report
The title still doesn't look good. I would propose: "Clinical and Research solutions to Manage Obstructive Sleep Apnea: A Review"
The manuscript has improved after the reviewers' comments.
The discussion still remains unbalanced towards HNS, consider discussing the results of this meta-analysis DOI: 10.1007/s11325-020-02069-2
Author Response
We would like to thank you for your comments to improve our paper further. We have revised our manuscript to answer your comments one by one in the attachment.

Reviewer 3 Report
Authors adressed my comments and improved the manuscript.
Author Response
We would like to thank you for your comments to improve our paper further. We have revised our manuscript to answer your comments in the attachment.
